# New Concept and Apparatus for Cytocentrifugation and Cell Processing for Microscopy Analysis

**DOI:** 10.3390/ijms22137098

**Published:** 2021-07-01

**Authors:** Anna Ligasová, Karel Koberna

**Affiliations:** Institute of Molecular and Translational Medicine, Faculty of Medicine and Dentistry and Czech Advanced Technology and Research Institute, Palacký University Olomouc, Hněvotínská 5, 779 00 Olomouc, Czech Republic

**Keywords:** cytocentrifugation, sample processing, staining, microscopy

## Abstract

Cytocentrifugation is a common technique for the capture of cells on microscopic slides. It usually requires a special cytocentrifuge or cytorotor and cassettes. In the study presented here, we tested the new concept of cytocentrifugation based on the threaded connection of the lid and the sample holder to ensure an adjustable flow of solutions through the filters and the collection of the filtered solutions in the reservoir during centrifugation. To test this concept, we developed a device for the preparation of cell samples on circular coverslips. The device was tested for the capture and sample processing of both eukaryotic and prokaryotic cells, cell nuclei, and mitochondria for microscopy analysis including image cytometry. Moreover, an efficient procedure was developed for capturing formaldehyde-fixed cells on non-treated coverslips without cell drying. The results showed that the tested arrangement enables the effective capture and processing of all of the tested samples and the developed device represents an inexpensive alternative to common cytocentrifuges, as only the paper filter is consumed during sample processing, and no special centrifuge, cytorotor, or cassette is necessary. As no additional system of solution removal is required during sample staining, the tested concept also facilitates the eventual automation of the staining procedure.

## 1. Introduction

The principle of the cytocentrifugation technique (also known as cytospin), serving for the attachment and concentration of cells obtained from biological fluids, was already formulated in 1963, when Bots and his colleagues put the cellular suspension in a narrow vertical container and allowed the cells to sediment onto the microscopic slides [1]. In 1965, Doré and Balfour presented an apparatus based on the above-mentioned principle; however, in contrast with it, they sped up the sedimentation by employing centrifugal force [2]. The first models of cytocentrifuges were presented by Dr. Watson in his paper [3]. Until then, the preparation of cell suspensions on microscopic slides was done by the direct smear technique of cell sediments prepared by conventional centrifugation. The smear technique usually resulted in the spreading of cells over a relatively large area. In samples with a low cell number, this makes the inspection of such slides difficult [3,4]. Although cytological specimens may be also deposited onto slides by other techniques such as touch preps or filter techniques [4,5], the cytocentrifugation technique has become the dominant approach for capturing cells from various samples in most laboratories, including clinical workplaces. 

Most present-day cytocentrifuges are based on the application of samples into special cassettes. These cassettes enable the mounting of the microscopic slide and also accommodate the filter, which serves to capture liquid during the centrifugation step. The cassettes also define the location of the slide towards the sample chamber and, along with the cassette holder, assure the correct direction of the centrifugal forces. After applying the sample in the chamber and the subsequent centrifugation step, the cells remain attached on the microscopic slide. Thanks to the possibility of changing the area of the attachment, it is possible to reach a relatively high increase in the concentration of captured cells, even when there is a low number of cells in the sample. 

In diagnostic laboratories, various biological fluids, such as urine, effusions (fluids from three body cavities—pleural, peritoneal, and pericardial [6]), blood, cerebrospinal fluid, synovial fluid, fine needle aspirates, and other samples of biological fluids containing cells, are processed by cytocentrifugation. The primary prerequisite of the preparation of samples for microscopy evaluation is a population of well separated fresh and intact cells [5]. After the cytocentrifugation step, these samples are commonly stained. This includes, e.g., staining according to the May−Grünwald−Giemsa protocol [7], Papanicolaou staining [8], or Gram staining of bacteria in the case of laboratories of clinical microbiology [9]. The sample can then be alternatively immunostained by antibodies raised against specific cellular components, or specific DNA or RNA sequences can be detected by fluorescence in situ hybridization (FISH) [10]. The method of cytocentrifugation is used also in veterinary medicine, for example for the assessment of low cell fluids such as cerebrospinal fluid, bronchoalveolar lavage fluid, or urine. [11]. 

Although cytocentrifugation using a special cytocentrifuge accelerates and improves the sample quality, it also exhibits some disadvantages, as follows:(1)Samples have to be typically removed from the special cassette before the staining step(s), the personnel working with the biological material can come into direct contact with the unfixed material.(2)Relatively large slides are used and the slides are commonly stained by immersion in the staining solution. Therefore, a large amount of staining solutions is necessary. Alternatively, special devices or a special approach has to be used, which commonly results in the prolongation of the procedure, making it more expensive.(3)A special cytocentrifuge or cytorotor is usually needed.(4)Relatively high expenses are necessary for the purchase of the cassettes used for cytocentrifugation. Consequently, it often leads to the repeated usage of these cassettes.

Because of the quite high expenses for the consumable supplies, other methods for the application of cells on the slides have been developed. An example is a method where the cells are concentrated by centrifugation using a classic centrifuge (at ca 80× *g*) and, subsequently, the cells are transferred using a plastic Pasteur pipette onto the slide. Then, using a so-called glass spreader, which is a tool made from a glass pipette, the cells are spread in a thin layer [12]. In addition, for example, Krishnamurthy and colleagues tested the use of custom-made filter cards for cytocentrifugation and compared them with those commercially available in order to reduce the high cost of these consumable supplies. They then compared the cell densities and found that the usage of 300 gsm handmade filter cards was just as effective as employing commercial filter cards [13].

In the study presented here, we proposed the concept of cytocentrifugation based on the threaded connection of two parts in order to ensure an adjustable flow of solutions through the filters and the collection of the filtered solutions in the reservoir during the centrifugation step. Based on this concept, we developed and tested an apparatus (called a CytoTrap) equipped with a threaded lid and a holder of the capturing carriers that make it possible to adjust the pressure on the filters. Depending on the arrangement of the CytoTrap, dry or wet samples can be obtained. The device was equipped with a reservoir in the form of common 50-mL centrifugation tubes for spilled solutions. This arrangement made it possible to capture and process the samples without the need to remove the capturing carrier from the device before processing the sample. It allowed for the use of common centrifuges equipped with a swing-out rotor for common 50 mL tubes. In addition, the CytoTrap could be used repeatedly. In this respect, the only consumables were the filters. Circular glass coverslips with a diameter of 12 mm were successfully tested as capturing carriers. These carriers are highly popular as they enable the performance of, e.g., immunocytochemical labeling using tiny drops of solutions (around 20 µL). The results showed that both the concept and the developed device are attractive alternatives for capturing and processing various samples, as its use resulted in both a high effectivity and the saving of time and cost.

## 2. Results and Discussion

### 2.1. Description of the Device

Depending on the purpose of the experiment, two basic arrangements of the CytoTrap are possible. The arrangement convenient for the preparation of dry samples is shown in Figure 1. In this case, the CytoTrap is comprised of the holder, the support for the glass coverslips, the filter, the insert, the rubber seal, and the lid. We refer to this arrangement as the dry arrangement or dry centrifugation. 

The holder is composed of two parts: the sample part and the collecting part. The sample part contains a bottom with an opening and provides the space for the support, coverslip, filter, insert, and the rubber seal (Figure 2a). The collecting part facilitates handling and also contains the opening, which serves for the drainage of the solutions filtrated during the processing of the cell suspension. This part was also successfully tested as a connector to vacuum pumps for solution removal (Figure 2b). The opening in the lid is used for filling the CytoTrap with the cell suspension and solutions. This opening can also be used for solution removal (Figure 2c). 

The lid and the holder are equipped with a threading. The threading serves as the mutual connection of these two parts and to create pressure on the seal, insert, filter, coverslip, and support. 

The lid also contains the recess serving for fixation of the CytoTrap on the neck of the 50-mL centrifugation tube (Figure 2d). The holder is equipped with two grooves (Figure 2a). One of the grooves passes through the whole wall of the holder. These grooves have two functions, namely: to fix the insert and to prevent the partial turning of the insert when tightening the lid, and to make manipulation with the support and coverslips easier. 

The insert is furnished with an opening. The opening serves as a container for solvents and/or mixtures that are used during sample processing (Figure 3a). The diameter of this opening and the length of the insert determines the maximum amount of solution that can be processed. Its neck is furnished with a groove that makes handling easier. The outer part of the insert is equipped with two protrusions (Figure 3a). They fit into the complementary grooves in the sample part of the holder. They prevent its slippage when tightening the lid that can result in distortion of the filter and coverslip breakage. The opening in the base that is closer to the bottom of the sample part of the holder defines the position where the particles will sediment on the coverslip or other sample carrier. The opening in the insert can be closed by the cap (Appendix A) during centrifugation.

The insert is pressed onto the other inner parts of the device (e.g., support, coverslip, filter, and seal) by the threaded connection between the lid and the holder. The support is put under the coverslip. It is furnished with a recess and a small hole on the upper side (Figure 3b). The depth of the recess is the same as the coverslip thickness or slightly deeper. For coverslips with a declared thickness of 0.2 mm, we use the recess with a depth of 0.25 mm. This recess serves for the positioning of the coverslip in the central part of the device. The hole facilitates manipulation with the coverslips. The support is also furnished with a system of grooves on the bottom side (Figure 3c). They enable drainage of the solutions from the processed sample. The support is optimized for coverslips with a diameter of 12 mm. The surface of the support has to be completely smooth to prevent coverslip breakage. During the assembly of the CytoTrap, the support with the coverslip is put into the sample part of the holder. Then, the filter is put on the support with the coverslip, followed by the insert and the seal. Finally, the lid is screwed on the holder (Appendix A). The procedure of the CytoTrap assembly is also shown in Appendix A. The disassembly is performed in the opposite direction.

After assembly, the CytoTrap is put on the neck of the 50-mL centrifuge tube. It can be done inside or outside of the centrifuge. The filling of the CytoTrap with the cell suspension can be done inside or outside of the centrifuge as well. After filling the device with the samples, the cell suspension is centrifuged in a centrifuge with a swing-out rotor (Appendix A). During this step, the cells are pushed against the coverslip and the solution is filtered and runs out (Figure 4a).

The solution runs through the openings in the collecting part of the holder and finally is collected in the 50-mL centrifugation tube (reservoir). The CytoTrap can be disassembled and the coverslips with cells can be removed or the CytoTrap can be used for subsequent cell processing, such as washing, fixation, or staining. The capacity of the reservoir is sufficient for more than 50 steps with 0.5 mL of solutions if the processing is performed in the centrifuge. Alternatively, the CytoTrap can be removed from the centrifuge and the cell processing can be performed by pipetting, when the pipette is used both for the filling and aspiration of the solutions. The solutions can also be removed by a vacuum, as the lower opening in the collecting part of the holder can serve as the connector of the vacuum part.

If drying of samples should be avoided, the seal in the form of a hollow cylinder is inserted between the coverslip and the filter (Figure 4b). The hollow cylinder serves as a wet chamber protecting the sample from drying during centrifugation. We refer to this arrangement as the wet arrangement or wet centrifugation.

The holder, the support, the insert, the lid, and the cap are made from polytetrafluoroethylene (PTFE) and can be therefore treated in a wide temperature range (−100 °C–+250 °C). As the seals are made from silicone rubber, they can be treated in a temperature range from −60 °C to +230 °C. In this respect, all of these parts can be autoclaved if the decontamination of the device is necessary.

### 2.2. Examples of the CytoTrap Use

We tested the CytoTrap for various samples and applications. It included the preparation and, in chosen cases, also the additional processing of the human cells, cell nuclei, mitochondria, and bacterial cells. 

We found that drying provided a very efficient method of sample fixation to the coverslips in all of the tested cases. On the other hand, as drying results in protein denaturation, it is less suitable for the detection of some cellular components by antibodies. In this respect, formaldehyde fixation and detergent permeabilization of the samples is usually recommended for the detection of many cellular proteins [14]. Although the antibody staining can be performed before cell capture, it is less convenient, as the staining protocol commonly involves several washing and incubation steps commonly resulting in the gradual loss of cells. This is especially critical if low-concentration cell samples are used. 

In the case of the wet centrifugation of human cells, but not of bacterial cells, we usually observed a much lower cell density than in the case of the dry samples. Our further experiments showed that this can be partially overcome by centrifugation of the cells on coverslips coated with gelatin or poly-lysine, followed by fixation in formaldehyde. 

The subsequent experiments showed that even higher cell capture efficiency was observed if an incubation with copper ions was performed after the formaldehyde fixation of the cells (Figure 5a). We found that the treatment of the captured cells on non-coated coverslips with 20 mM copper sulphate shortly after the formaldehyde fixation is sufficient for stabilization of cells if the wet arrangement is used. Such treatment resulted into around an 11.6 ± 3.2-fold increase of the efficiency of the cell capture when compared with the non-treated cells. The capture efficiency was similar to the dry arrangement (Figure 5b). Although copper is a known cofactor for crosslinking, e.g., elastic fibers or collagen in connective tissues [15], it is not clear if this mechanism can participate in the stabilization of cells on glass coverslips. Importantly, no effect of copper ions was observed if ethanol-fixed cells were captured by wet centrifugation. As copper(II) was found to oxidize formaldehyde through the formation of copper(I) [16], we tested the utilization of copper(I) to increase the capture efficiency of formaldehyde- or ethanol-fixed cells. In this respect, we found that copper(I) increases the capture efficiency of formaldehyde-fixed cells as well. However, no effect was observed if ethanol-fixed cells were used. This indicated that the observed effect depends on the presence of both formaldehyde-fixed cells and copper ions. 

The next set of experiments also showed that in samples with a low cell density, it is more suitable to perform the whole procedure of cell processing (e.g., permeabilization, washing, and staining) in the device and to use the centrifugal forces for the removal of the solution. 

We also compared the samples prepared by CytoTrap with the samples prepared by the conventional cytocentrifuge. In these experiments, we used CCRF-CEM suspension cells. The area available for the cell capture was around 49 mm^2^ in the case of the used cytocentrifuge cassette and around 44 mm^2^ in the case of the CytoTrap. Around 50,000 cells were centrifuged per sample. After centrifugation, slides (conventional cytocentrifuge)/coverslips (CytoTrap) were removed from the cassette/holder, mounted in the mounting medium, and observed. The results showed that the cell capture efficiency is very similar in both cases (Figure 5c). 

#### 2.2.1. DNA Replication and Cell Cycle Analysis Using the CytoTrap

To test the possibility to use CytoTrap for sample staining, we prepared wet and dry samples of HeLa S3 cells labeled by a short EdU pulse. After fixation with ethanol (dry centrifugation) or with formaldehyde (wet centrifugation), the cells were stained with 6-FAM azide using the click reaction, followed by DAPI staining (Figure 6). All staining and washing steps were performed in the CytoTrap, and centrifugation was used for the removal of particular solutions. In both cases, we observed similar signals corresponding to overall DNA (DAPI signal, Figure 6) and replicated DNA (EdU signal, Figure 6). 

Similar results were obtained if the samples were removed from the CytoTrap after fixation followed either by cell drying in the case of ethanol-fixed cells or after washing with 1× PBS after fixation with formaldehyde and permeabilization with Triton X-100. The removal of the coverslips from the CytoTrap and sample labeling performed on the drops of solutions resulted in the lowering of the amount of solutions used, as the minimum volume of the solutions necessary was 100 µL in the case of CytoTrap and around 20 µL in the case of the drops on a parafilm sheet.

Furthermore, we prepared the samples for subsequent cell cycle analysis using CytoTrap and the wet arrangement. The BrdU-labeled cells were fixed and permeabilized in CytoTrap. After washing, coverslips were removed from the CytoTrap and were incubated in low concentration HCl. Then, the samples were incubated on the drops of the primary anti-BrdU antibody (20 µL) and subsequently on the drops of the secondary antibody conjugated with Alexa Fluor 488 fluorochrome and DAPI. The image cytometry was used to analyze the cell cycle (Figure 7). The DNA histogram (Figure 7a), bivariate plot of the DNA and replication signal (Figure 7b), and the estimated lengths of the particular phases of the cell cycle (Figure 7c) were evaluated. The performed cell cycle analyses of HeLa S3 cells labeled with BrdU showed that the CytoTrap can be used also for the sample processing of such types of analyses, without any destruction of cells or without any impact on the signals.

#### 2.2.2. Giemsa−Romanowski Staining Using the CytoTrap

The CytoTrap was also tested for the preparation of samples stained with the highly popular Giemsa−Romanowski staining (Figure 8a). This staining is very often used in diagnostic laboratories. In these laboratories, most of the tested samples come from the various biological fluids that contain more or less number of cells. Depending on the type of diagnostic test, the first step is the cytocentrifugation and, if necessary, the concentration of the cells of interest on the microscopy slide followed by staining. The whole procedure, namely cell capture and staining, was performed in CytoTrap. Centrifugation was used for the removal of solutions. We used suspension HeLa S3 cells in these experiments. From the Figure 8a, it is clear that the cells are efficiently stained both by Giemsa−Romanowski and DAPI staining.

#### 2.2.3. Another Possible Usage of the CytoTrap

Finally, CytoTrap was successfully tested for the preparation of nuclear DNA halos (Figure 8b), for the preparation of samples of isolated human mitochondria (Figure 8c), and for the preparation of samples with bacteria on clean coverslips (Figure 9) or on coverslips covered with human adherent cells infected with mycoplasmas (see Supplementary Figure S5 in [17]).

The DNA halo is a technique usually used for the measurement of DNA damage caused by various agents. The name halo is derived from the shape of radially extracted DNA fragments from isolated nuclei [18]. During the procedure, isolated cell nuclei have to be attached to the slides and cytocentrifugation is often used [19]. In these experiments, isolated cell nuclei suspension prepared from adherent HeLa cells were centrifuged by wet centrifugation using CytoTrap. The rest of the steps concerning the DNA halo preparation were performed outside of CytoTrap. An example of the prepared nuclear DNA halo is shown in Figure 8b. It is comparable with those prepared by common cytocentrifuge (e.g., in [19]). 

The mitochondria were isolated from adherent IMR-90 cells and subsequently attached to the coverslips by wet centrifugation. After fixation and washing, the samples were removed from the CytoTrap and antibody staining was done on the parafilm sheet. The mitochondria stained with the mitochondrial marker were observed only in the experimental sample. No signal was observed in the control sample without mitochondria (Figure 8c).

We deposited two bacterial strains, *E. coli* and *S. epidermidis* (Figure 9), on clean glass coverslips. For the deposition, we used wet centrifugation. The fixation and washing steps were performed using CytoTrap. The permeabilization and DAPI labeling steps were done on drops of the solutions on the parafilm sheet. 

We also used CytoTrap for capturing bacterial cells on coverslips with adherent cells (see Supplementary Figure S5 in [17]). In this case, the Lep cells accidentally infected with the mycoplasma were grown on the coverslips. The bacteria *E. coli* were deposited on the coverslips by wet centrifugation. The fixation and washing steps were performed in the CytoTrap. The rest of the procedure was performed on the drops of solutions [17].

These results clearly showed that the presented concept and the developed device could efficiently serve for the concentration, attachment, and additional processing of various cells or cell organelles. In this respect, CytoTrap can supplement the currently used cytocentrifugation techniques as it provides an easy and inexpensive way of preparing various samples for microscopy analysis.

## 3. Materials and Methods

### 3.1. Cell Cultures and Bacteria Cells

The suspension HeLa S3 cells (human cervix, adenocarcinoma, a gift from Dr. David Staněk, Institute of Molecular Genetics CAS, Prague) were cultivated in Minimum Essential Medium Eagle, Spinner modification (S-MEM, Sigma Aldrich, St. Louis, MO, USA) supplemented with 10% fetal bovine serum (Gibco, ThermoFisher Scientific, Waltham, MA, USA), 3% L-glutamine (Sigma Aldrich, St. Louis, MO, USA) and 50 μg/mL of gentamicin (Lek Pharmaceuticals, Ljubljana, Slovenia). The adherent IMR-90 cells (human diploid fibroblasts, lung, ATCC, Manassas, VA, USA, CCL-186) were cultivated in Eagle’s minimum essential medium (EMEM, Sigma Aldrich, St. Louis, MO, USA) supplemented with 20% fetal bovine serum (Gibco, ThermoFisher Scientific, Waltham, MA, USA), 3.7 g/L of sodium bicarbonate (Sigma Aldrich, St. Louis, MO, USA), and 50 µg/mL of gentamicin (Lek Pharmaceuticals, Ljubljana, Slovenia). The adherent HeLa cells (human cervix, adenocarcinoma; ATCC, Manassas, VA, USA, CCL-2) were cultivated in Dulbecco’s modified Eagle’s medium (DMEM, Gibco, ThermoFisher Scientific, Waltham, MA, USA) supplemented with 10% fetal bovine serum (Gibco, ThermoFisher Scientific, Waltham, MA, USA), 3.7 g/L of sodium bicarbonate (Sigma Aldrich, St. Louis, MO, USA), and 50 µg/mL of gentamicin (Lek Pharmaceuticals, Ljubljana, Slovenia). The suspension CCRF-CEM cells (peripheral blood, acute lymphoblastic leukemia, ATCC, Manassas, VA, USA, CCL-119) were cultivated in an RPMI-1640 medium (Lonza Biosciences, Basel, Switzerland) supplemented with 10% fetal bovine serum (Gibco, ThermoFisher Scientific, Waltham, MA, USA) and 100× diluted solution of penicillin and streptomycin (Gibco, ThermoFisher Scientific, Waltham, MA, USA). The cells were cultivated at 37 °C in a humidified atmosphere containing 5% CO_2_. The cell lines were regularly tested for mycoplasma contamination by PCR and enzymatic detection [17,20].

The bacteria cells were obtained from the culture collection of the Department of Microbiology, Faculty of Medicine and Dentistry, Palacký University Olomouc, where they were cultivated in the Mueller Hinton broth (BIORAD, Hercules, CA, USA) for 16 h at 37 °C. *Escherichia coli* (ATCC, Manassas, VA, USA, 25922, *E. coli*) and *Staphylococcus epidermidis* (CCM 7221, *S. epidermidis*; the culture collection of the Department of Microbiology, Faculty of Medicine and Dentistry, Palacký University Olomouc) were used [17]. 

### 3.2. Preparation of Cytospin Samples

Circular glass coverslips with a diameter of 12 mm were used for the cell capture. For the preparation of the dry samples, the circular filter was placed on the glass coverslip. For the preparation of the wet samples, the circular rubber seal was placed between the filter and the coverslip (http://transfer.vtpup.cz/cytotrap/?lang=en, accessed on 21 May 2021). The diameter of the circular opening in the filter and the seal was 10 or 7.5 mm. The filter was cut from chromatography paper (Whatman, cat no: 3030-6188). The width of the seal was 2 mm. 

#### 3.2.1. Preparation of Fixed Cell Samples

The HeLa S3 cells were fixed by 2% formaldehyde in a 1× PBS buffer or by 70% ice-cold ethanol. Fixation was performed before or after capturing the cells on the coverslips. 

If the formaldehyde fixation was used before cell capture, the cells were transferred from the culture flask to the 15-mL tube (10 mL) and were centrifuged at 50× *g* for 5 min. Then, the culture medium was removed, replaced by 1× PBS, and the samples were centrifuged at 50× *g* for 5 min. The buffer was replaced by one volume of 1× PBS, and the pellet was disturbed by repeated pipetting. Then, nine volumes of 2% formaldehyde were added and the cells were incubated on the laboratory shaker for 10 min at 150× *g*. 

If ethanol fixation was used, 1× PBS was removed and 1 mL of 70% ice-cold ethanol was added instead of formaldehyde and the cells were fixed for 1 h at −20 °C. The cells were left in ethanol until they were applied to the device. 

#### 3.2.2. Preparation of Non-Fixed Cells by the Dry Centrifugation

If the non-fixed cell samples were prepared by dry centrifugation, the device was assembled as shown in Figure 1. In this case, 0.05 mL of the cell suspension in a growth medium (ca 7 × 10^4^ of cells) was applied into the device prefilled with 0.5 mL of 150 mM NaCl and 5% BSA and centrifuged at 150× *g* for 10 min.

#### 3.2.3. Preparation of Ethanol-Fixed Samples by Dry Centrifugation

The device was assembled as shown in Figure 1. Then, 0.05 mL of previously ethanol-fixed cells (see chapter 3.2.1; ca 7 × 10^4^ of cells) was applied into the device prefilled with 0.5 mL of 150 mM NaCl and centrifuged at 150× *g* for 10 min. Alternatively, non-fixed samples, prepared as described in Chapter 3.2.2, were incubated for 10 min with 0.5 mL of the ice-cold 70% ethanol at −20 °C or at room temperature (RT), and were subsequently centrifuged at 150× *g* for 10 min at RT.

#### 3.2.4. Preparation of Formaldehyde-Fixed Samples Using Wet Centrifugation

In this case, the device was assembled as shown in Figure 4b. Then, 0.5 mL of formaldehyde-fixed cells prepared as described in Chapter 3.2.1 (ca 7 × 10^4^ of cells) was applied into the device and centrifuged at 150× *g* for 5 min at RT. Alternatively, 0.05 mL of non-fixed cells (see Chapter 3.2.2; ca 7 × 10^4^ of cells)) in 1× PBS was applied to the device containing 0.5 mL 2% formaldehyde and was incubated for 5 min. The samples were then centrifuged at 150× *g* for 10 min at RT. Then, 0.5 mL 150 mM NaCl was added and the samples were centrifuged again at 150× *g* for 5 min at RT. After centrifugation, a solution of 20 mM CuSO_4_ and 150 mM NaCl (0.05 mL) was added and the samples were centrifuged at 150× *g* for 10 min at RT. In some cases, instead of the solution of 20 mM CuSO_4_ and 150 mM NaCl, we used a solution of 20 mM CuSO_4_, 40 mM sodium ascorbate, and 150 mM NaCl (copper(I) solution). 

### 3.3. Preparation of Bacteria Samples by Wet Centrifugation

The device was assembled as shown in Figure 4b. Then, 0.02 mL of *E. coli* or *S. epidermidis* was applied to the device with 0.5 mL of 1× PBS and centrifuged (150× *g*, 5 min, RT). Subsequently, 0.5 mL of 2% formaldehyde in 1× PBS was added. After a 5-min incubation, the samples were centrifuged (150× *g*, 5 min, RT), washed with 1× PBS, and centrifuged again (150× *g*, 5 min, RT). Next, the samples were removed from the device and the cells were permeabilized with 0.2% Triton X-100 in 1× PBS (10 min, RT). Then, the samples were stained using the enzymatic labeling and with 10 µM DAPI (30 min, RT) [17]. 

### 3.4. Preparation of DNA Halo Samples

The developed device was also tested for the preparation of cell nuclei for DNA halo analysis. The modified protocol described in [19] was used. Briefly, the adherent HeLa cells were washed with 1× PBS, removed from the bottom of the culture flask by a cell scraper, and centrifuged for 10 min at 6000× *g* at 4 °C. After buffer removal, the cells were incubated with the nuclei buffer (10 mM Tris-HCl, pH 8; 3 mM MgCl_2_; 0.1 M NaCl; 0.3 M sucrose; protease inhibitors) with 0.5% Nonidet P40 (1 h, on ice). The samples were homogenized by a Dounce homogenizer (ca 100×). The samples were transferred into the developed device arranged for wet centrifugation (see Figure 4b) and were centrifuged for 10 min at 300× *g*. Then, the samples were removed from the device, shortly immersed in a buffer containing 25 mM Tris-HCl, pH 8; 0. 2 mM MgCl_2_; 0.5 M NaCl; 1 mM PMSF and protease inhibitors. The samples were further incubated in the halo buffer (10 mM Tris-HCl, pH 8; 1 mM DTT; 2 M NaCl; 10 mM EDTA; protease inhibitors) for 4 min. The samples were subsequently washed in buffer A (25 mM Tris-HCl, pH 8; 0.2 mM MgCl_2_; 0.2 M NaCl) and in buffer B (25 mM Tris-HCl, pH 8; 0.2 mM MgCl_2_), fixed with 2% formaldehyde, stained with 10 µM DAPI, and mounted in glycerol.

### 3.5. Isolated Mitochondria

The isolated mitochondria were captured on coverslips using the developed device and the arrangement for wet centrifugation (see Figure 4b). The adapted protocol of Frezza and colleagues [21] was used for the isolation of the mitochondria. Adherent IMR-90 cells were cultivated in the Petri dishes. After medium removal, the cells were washed with the ice-cold 1× PBS, detached by a cell scraper, and the cell suspension was transferred into the Eppendorf tube and centrifuged 10 min at 600× *g* at 4 °C. The supernatant was removed, and the pellet was resuspended in IBc buffer (10 mM Tris-MOPS, pH 7.4; 1 mM EGTA-Tris, pH 7.4; 20 mM sucrose; protease inhibitors). The suspension was transferred into the Dounce homogenizer and the cells were homogenized 40×. Furthermore, the homogenate was centrifuged 10 min at 600× *g* at 4 °C. The supernatant was then transferred into a new Eppendorf tube and centrifuged 10 min at 7000× *g* at 4 °C. The supernatant was discarded, 1× PBS was added, the pellet was resuspended, and the suspension was centrifuged 10 min at 7000× *g* at 4 °C. The buffer was removed, 2% formaldehyde was added, and the pellet was resuspended and incubated for 10 min at RT. Then, the suspension was transferred into the developed device arranged for wet centrifugation and centrifuged for 10 min at 3000× *g* at RT. After centrifugation, 1× PBS was added, the samples were centrifuged for 10 min at 3000× *g* at RT. The samples were then transferred from the device onto the drops of the 1× PBS followed by permeabilization in 0.2% Triton X-100 in 1× PBS (10 min, RT). After washing, the samples were incubated with the anti MTCO-2 primary antibody (Abcam, 1:100, 30 min, RT) and the secondary antibody conjugated with Alexa Fluor488 (Jackson ImmunoResearch, 1:100, 30 min, RT).

### 3.6. Sample Preparation by the Conventional Cytocentrifuge

CCRF-CEM cells were diluted in 1× PBS to a concentration of 50,000 cells per ml. Then, 0.5 mL of the cell suspension was transferred to the chambers composed of the adaptor, microscopic slide, and filter paper with the square cut. The chambers were put into the cytocentrifuge (Cyto-Tek, Sakura, Alphen aan den Rijn, The Netherlands) and centrifuged at 29× *g* for 6 min. Then, the slides were removed from the chambers, covered with the mounting medium and coverslip, and were observed.

### 3.7. Staining Protocols

If not stated otherwise, the particular steps were performed in the developed device and the centrifugation was used for the solution removal (150× *g*, 5 min, RT). Alternatively, the staining and washing solutions were removed by pipette or vacuum pump. The vacuum pump was connected to the lower opening in the collecting part of the holder (Figure 2b). In the indicated cases, the coverslips were removed from the device and the particular incubation steps were performed on drops of the solutions.

#### 3.7.1. DAPI Staining

The captured samples were incubated with a solution of 10 µM DAPI in 20 mM Tris-HCl, pH 7.4 and 150 mM NaCl for 30 min at RT. Then, the samples were washed with a buffer composed of 20 mM Tris-HCl, pH 7.4, and 150 mM NaCl. 

#### 3.7.2. Giemsa−Romanowski Staining

For Giemsa−Romanowski staining, HeLa S3 cells were used. In this case, 500 µL of 10% BSA in deionized water was pipetted into the developed device. Then, 100 µL of the cell suspension was added and the cells were centrifuged for 20 min at 150× *g* and room temperature. Then, 100% methanol was added and the samples were incubated for 10 min. After methanol removal, the samples were incubated with May−Grünwald solution for 3 min. After solution removal, the samples were incubated with Giemsa−Romanowski staining solution for 8 min and were washed three times with deionized water. In some experiments, the samples were co-stained with 10 µM DAPI.

#### 3.7.3. Labeling of DNA Replication by EdU

For simultaneous EdU and DAPI labeling, we used adapted protocol from [22,23]. In these experiments, the cells were incubated with 10 µM EdU for 30 min. Then, the ethanol-fixed or formaldehyde-fixed and Triton X-100-permeabilized HeLa S3 cells were first incubated in a click solution containing Alexa Fluor 488 azide (30 min, RT) and after washing in 1× PBS stained with DAPI. In some cases, the samples were removed from the device after the fixation step and the following incubations were done on drops of solutions.

#### 3.7.4. Labeling of DNA Replication by BrdU

For the labeling of cells by BrdU and DAPI, the adapted protocol from [24] was used. Briefly, the HeLa S3 cells were first incubated with BrdU for 30 min. Then, the samples were processed according to the adapted protocol described in Chapters 3.2.1 and 3.2.4. The samples were centrifuged for 5 min at 150× *g*. After washing in 150 mM NaCl, the cells were permeabilized with 0.2% Triton X-100 in 1× PBS, washed with 1× PBS (2×) and 150 mM NaCl and 3 mM KCl (3×). During these steps, 0.5 mL of solution was added and the cells were centrifuged for 5 min at 150× *g*. The samples were removed from the device and the incorporated BrdU was revealed by incubation in a solution of 20 mM HCl in 150 mM NaCl and 3 mM KCl (20 min, 25 °C). The next steps were performed on the drops of solutions. After washing with 1× buffer for exonuclease III, the cells were incubated in a solution of the primary anti BrdU antibody (clone Bu20a, 5 µg/mL), 1× buffer for exonuclease III, and exonuclease III (0.4 U/µL). The samples were quickly washed with 1× PBS (less than 1 min); post-fixed with 0.2% formaldehyde (10 min, RT); washed with 25 mM Tris-HCl, pH 7.5, and 150 mM NaCl; and incubated with the secondary antibody (Alexa Fluor 488 anti-mouse, 1:100) and DAPI (10 µM) for 30 min at RT. In some cases, samples were removed from the device after the fixation step and the following incubations were done on drops of solutions.

### 3.8. Fluorescence Microscopy 

Most of the images were acquired using an Olympus IX83 microscope (UPLSAPO O objective 100×, NA 1.4 or UPLFLN 2PH 10×, NA 0.3 or LUCPLFLN PH 20×, NA 0.45) equipped with a Zyla camera (Andor) with a resolution of 2048 × 2048 or 1024 × 1024 pixels using acquisition software (CellSense Dimension, Olympus, Tokyo, Japan). Some of the images were acquired using an Olympus IX81 microscope (UPLSAPO O objective 100×, NA 1.4 or UPLFL PH objective 40×, NA 0.75) equipped with a Hamamatsu ORCA II camera with a resolution of 1344 × 1024 pixels using Cell∧R acquisition software (Olympus, Tokyo, Japan). In some cases, the images were acquired in the *Z* stack mode and the final images are presented as a projection of the maximal intensity of the *Z* stack [17,25].

### 3.9. Data Evaluation

The data concerning the cell cycle analysis and DNA replication signal were analyzed using CellProfiler [26,27], ilastik [28], and Microsoft Excel software. The final graphs were made in GraphPad Prism 6 (GraphPad Software, San Diego, CA, USA) and in Python (with the numpy, pandas, matplotlib, and seaborn modules). All of the measurements were performed for three independent experiments. 

## 4. Conclusions

We have proposed a new concept for the preparation of cytospin samples enabling concurrent sample processing within the same device. In this respect, we constructed and tested the modular apparatus according to the proposed concept. We showed that the tested concept produces a highly effective way of both preparing and processing prokaryotic and eukaryotic cells, cell nuclei, and mitochondria without the need for a special cytocentrifuge and/or special machine for sample staining. CytoTrap is fully compatible with the common centrifuges with the swing-out rotors, fits common 50-mL centrifuge tubes, and can be used both for the dry and wet procedures of sample preparation and processing.

We have also developed a protocol for the effective attachment of formaldehyde-fixed cells to the non-treated coverslips without the need for sample drying. In this respect, we showed that the incubation step with the copper ion solution significantly increases the attachment of formaldehyde-fixed cells to non-treated glass coverslips. 

## 5. Patents

Palacký University Olomouc holds a Czech patent (307415) for the method for the determination of mycoplasmas using enzymatic labeling, a Czech patent (307454), and pending European patent application (EP 17837846.9) for the device facilitating the handling of solvents, mixtures, and samples on carriers. Names of inventors: A.L. and K.K.

## Figures and Tables

**Figure 1 ijms-22-07098-f001:**
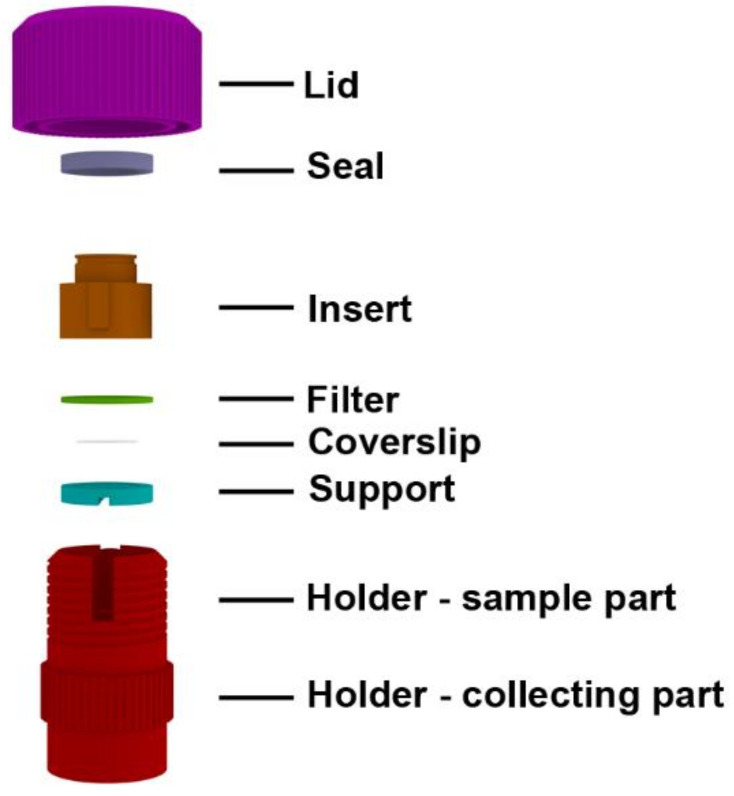
The schema of the developed device in the arrangement for dry centrifugation.

**Figure 2 ijms-22-07098-f002:**
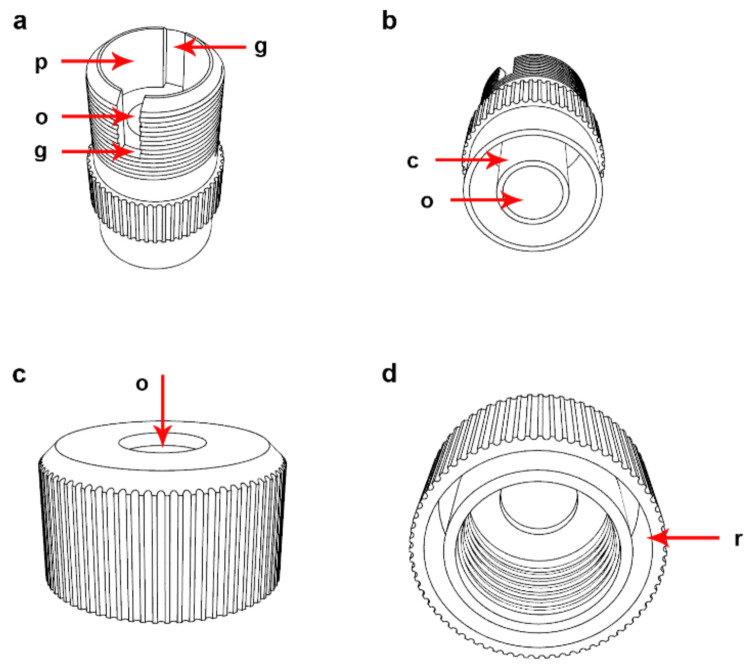
Description of the particular parts of the developed device. (**a**) The sample part of the holder. g—groove; o—opening at the bottom; p—place for the insertion of the support, coverslip, filter, insert, and the rubber seal. (**b**) The collection part of the holder. o—opening for drainage of solutions during processing of cell suspension; c—connector to vacuum pumps for solution removal. (**c**) The lid of the device. o—opening used for filling the device with cell suspension and for solutions filling and removal. (**d**) The lid of the device. r—recess for fixation of the device on the neck of the 50-mL centrifugation tube.

**Figure 3 ijms-22-07098-f003:**
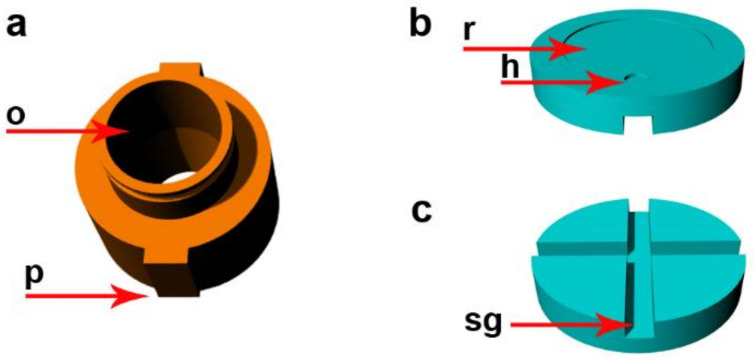
Description of the insert and support. (**a**) The insert. o—opening serving as a container for solvents and/or mixtures; p—protrusions complementary to the grooves in the sample part of the holder. (**b**) The upper view of the support. r—recess for accommodation of the coverslip; h—the small hole on the upper side to improve manipulation with the coverslip. (**c**) The bottom view of the support. sg—the system of grooves for the drainage of solutions.

**Figure 4 ijms-22-07098-f004:**
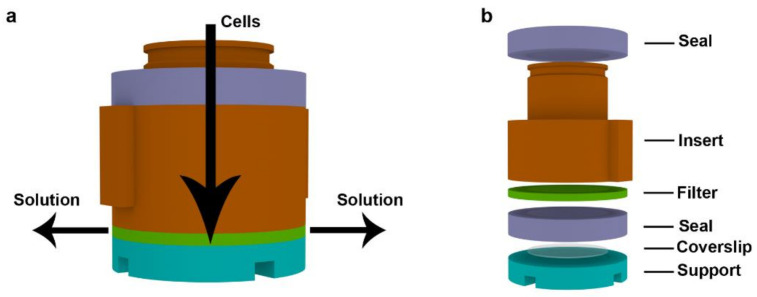
The schema of the flow of the filtered solution and the schema of the CytoTrap arrangement used for wet centrifugation. (**a**) The schema of the flow of the filtered solution through the filter (only the inner part of the device is depicted) during centrifugation is shown. The solution is collected in the 50 mL tube and the cells are captured on the surface of the coverslip. The arrows show the flow of the cells and solution during centrifugation. (**b**) The schema of the CytoTrap arrangement used for wet centrifugation.

**Figure 5 ijms-22-07098-f005:**
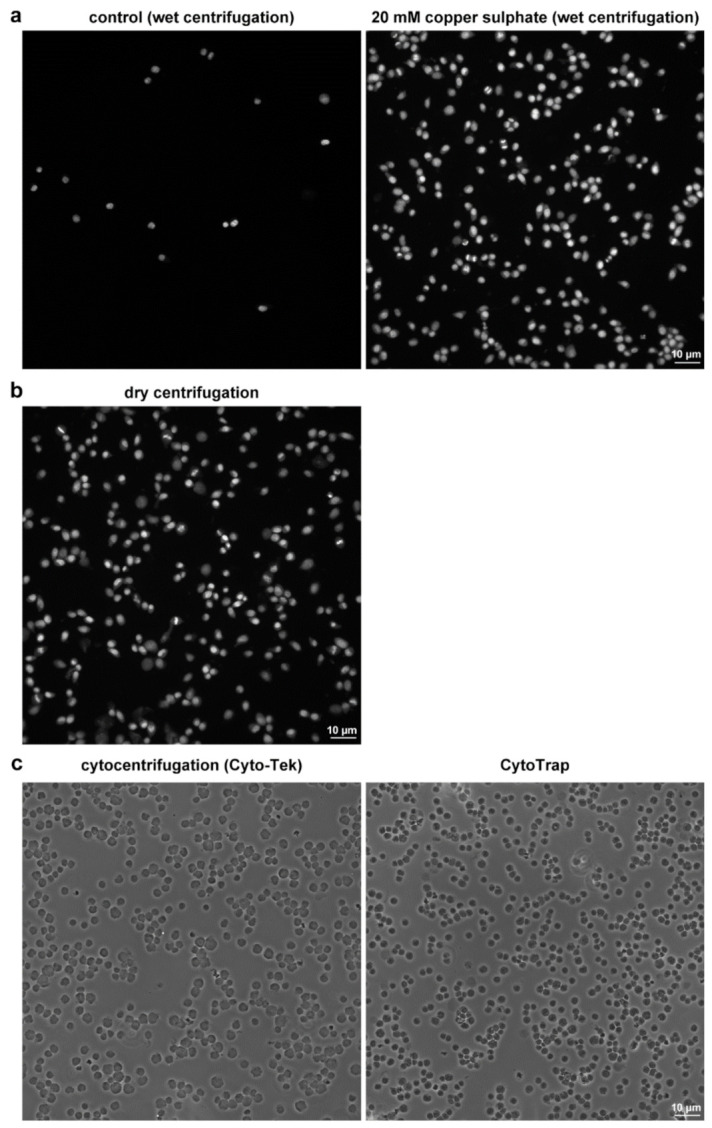
Comparison of the capture efficiency. (**a**) Comparison of the capture efficiency in samples of HeLa S3 cells prepared with the wet centrifugation and fixed by formaldehyde solution without (left) or with (right) the additional incubation of samples in the 20 mM copper sulphate solution. The whole procedure including DAPI staining was done in CytoTrap. (**b**) Example of the capture efficiency in samples processed by dry centrifugation and formaldehyde fixation. The whole procedure including DAPI staining was done in CytoTrap. The same cell concentration of the cell suspension was used in 5a and 5b. (**c**) Comparison of the samples prepared by conventional cytocentrifuge (left) and by CytoTrap (right). The same cell concentration of the cell suspension was used in both cases. Scale bar = 10 µm.

**Figure 6 ijms-22-07098-f006:**
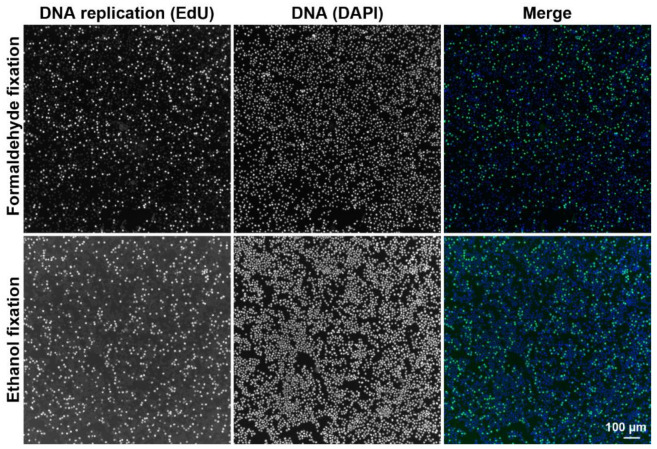
Example of the detection of DNA replication and DNA in HeLa S3 cells. The cells were labeled with EdU for 30 min and captured on glass coverslips. EdU was revealed by click reaction with 6-FAM azide (green) and the DNA was detected by DAPI (blue). The cells were processed either by wet centrifugation (formaldehyde fixation) or by dry centrifugation (ethanol fixation). Scale bar = 100 µm.

**Figure 7 ijms-22-07098-f007:**
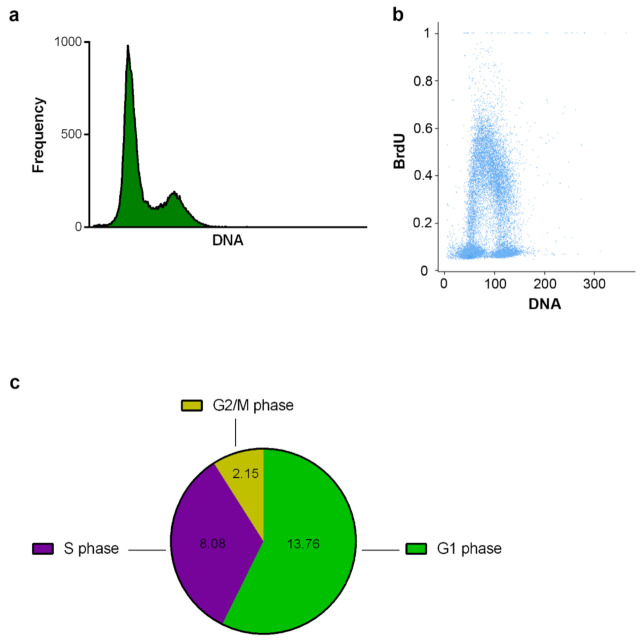
Cell cycle analysis. HeLa S3 cells were labeled with BrdU and captured on glass coverslips. The samples were evaluated by image cytometry after the immunostaining of BrdU and staining of DNA by DAPI. The DNA histogram (**a**), the bivariate plot of the DNA and BrdU signal (**b**) and the estimated length of the particular phases of the cell cycle in hours (**c**) are shown.

**Figure 8 ijms-22-07098-f008:**
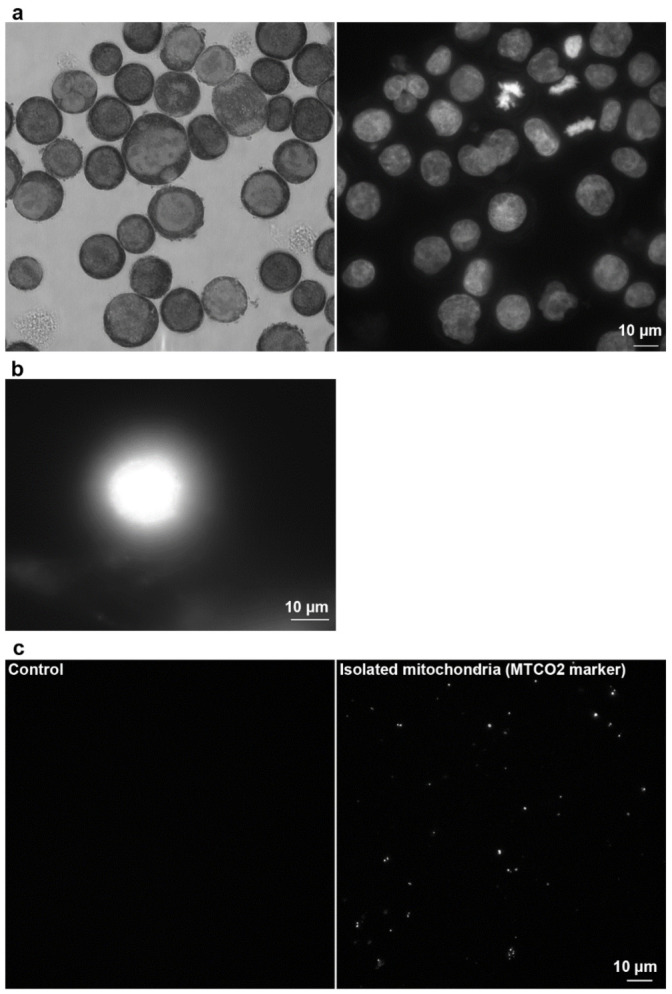
Examples of the use of the developed device for capturing and/or processing cells, cell nuclei, and mitochondria. (**a**) Giemsa−Romanowski (left) and DAPI (right) staining of HeLa S3 cells. (**b**) Nuclear DNA halo samples prepared from the nuclei of HeLa cells. (**c**) Captured and stained mitochondria of IMR-90 cells. The MTCO2 marker was used for mitochondria staining. The control sample did not contain mitochondria. Scale bar = 10 µm.

**Figure 9 ijms-22-07098-f009:**
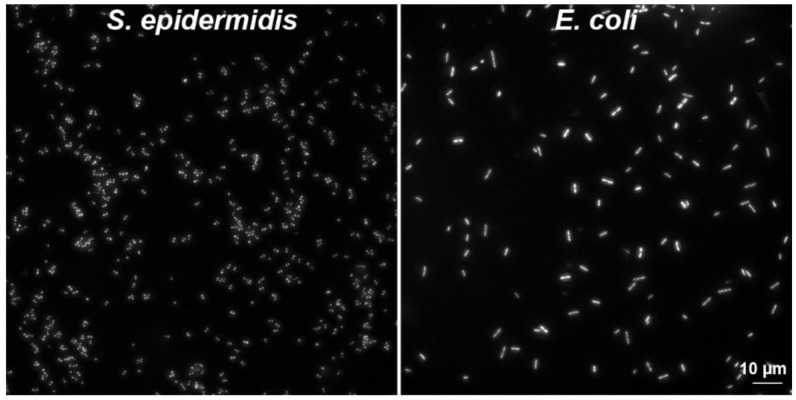
Examples of captured bacteria strains. *E. coli* and *S. epidermidis* were deposited on the glass coverslips by CytoTrap, fixed by formaldehyde, permeabilized by Triton X-100, and the bacterial DNA was stained by DAPI. Scale bar = 10 µm.

## Data Availability

All relevant data are included within the manuscript.

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
