# Peer review of "New Concept and Apparatus for Cytocentrifugation and Cell Processing for Microscopy Analysis"

_ijms, 2021, doi:10.3390/ijms22137098_

Round 1

Reviewer 1 Report

I quite like the idea of this manuscript, which is to use the process of cytocentrifugation to improve the accuracy and utility of microscopy analysis. Nonetheless, there are a few minor issues (and some less minor ones) that combine to reduce my overall enthusiasm for this manuscript, and will need to be addressed before I can recommend publication. These include:

  1. The authors list body fluids in the introduction. This term should be changed to “biological fluids” to reflect improved accuracy.
  2. Within this list, the term ‘effusions’ is unclear and should be clarified.
  3. The authors mention briefly that cytocentrifugation is also used in veterinary medicine but provide no additional details. They should either expand on this topic, at least a little, or consider deleting the sentence from the introduction.
  4. There are a variety of punctuation and syntax-based errors scattered throughout the manuscript that can distract from the ability to focus exclusively on the scientific content. The authors should consider how to best address this issue.
  5. I am also not sure how the term ‘cytocentrifugation’ is an accurate descriptor of the apparatus being described. The authors should consider explicitly defining how they arrived at this term.

More broadly, while the authors mention other options and currently used methods in the introduction, they do not provide a direct, explicit comparison between how their apparatus performs and how other approaches would perform. In the figures, for example, the authors should prepare some of these cells using conventional approaches and then the advantages of their current system would be apparent. As shown, this information is certainly interesting but it is hard to judge the actual utility of this set up without that comparison.

I recommend that the authors conduct experiments in which the same types of cells are prepared both by conventional methods and by their new approach, add that data and those figures to this manuscript, and then resubmit the manuscript for publication at that point in time.

Author Response

Thank you for all your valuable comments. Below find our replies.

1, The authors list body fluids in the introduction. This term should be changed to “biological fluids” to reflect improved accuracy.

We replaced the term „body fluids“ by the term „biological fluids“ throughout the whole manuscript.

2, Within this list, the term ‘effusions’ is unclear and should be clarified.

We included short explanation of the term effusion and related citation as well.

3, The authors mention briefly that cytocentrifugation is also used in veterinary medicine but provide no additional details. They should either expand on this topic, at least a little, or consider deleting the sentence from the introduction.

We added some examples of the use of the cytospin in the veterinary laboratories.

4, There are a variety of punctuation and syntax-based errors scattered throughout the manuscript that can distract from the ability to focus exclusively on the scientific content. The authors should consider how to best address this issue.

We sent the manuscript to our native English-speaking editor for additional grammar correction.

5, I am also not sure how the term ‘cytocentrifugation’ is an accurate descriptor of the apparatus being described. The authors should consider explicitly defining how they arrived at this term.

For the sake of simplicity and clarity, we replace the term cytocentrifugation in the case of our device by the term centrifugation.

6, More broadly, while the authors mention other options and currently used methods in the introduction, they do not provide a direct, explicit comparison between how their apparatus performs and how other approaches would perform. In the figures, for example, the authors should prepare some of these cells using conventional approaches and then the advantages of their current system would be apparent. As shown, this information is certainly interesting but it is hard to judge the actual utility of this set up without that comparison.

I recommend that the authors conduct experiments in which the same types of cells are prepared both by conventional methods and by their new approach, add that data and those figures to this manuscript, and then resubmit the manuscript for publication at that point in time.

We added a comparison of CCRF-CEM cells prepared by our device and by the common cytocentrifuge available at our institute. We added the text about it in the Material and methods part and in the Results and Disscusion part. We also added the new Figure 5c to this section.

Reviewer 2 Report

Anna Ligasová and Karel Koberna in their manuscript “New Concept and Apparatus for Cytocentrifugation and Cell Processing for Microscopy Analysis” introduce a device that allows depositing the suspension cells on the coverslip. The main advantage of this device over the similar cytocentrifugation techniques is that it is reusable and only requires cheap Whatman paper filters as consumables. Without any doubts, such device is of interest to scientific community. However, in my opinion, the manuscript itself requires substantial improvements before I would be able to recommend it for publishing.

Major comments

  1. Lines 204-208: “The subsequent experiments showed that even the higher efficiency of the cell capture was observed if an incubation with copper ions was performed after the formaldedyde fixation of cells. We found that the treatment of the captured cells on non-coated coverslips with 2-50 mM copper sulphate shortly after the formaldehyde fixation is sufficient for stabilization of cells if wet arrangement is used.”
    no data is provided to support this claim. Comparison of images with and without incubation with copper sulfate should be provided, quantification and statistics from repeated experiments are also required. Mechanisms leading to cell stabilization should be discussed.
  2. It seems that all parts of the device come in contact with the biological sample. Moreover, during the centrifugation no lid is used, which carries a danger for aerosol formation and spills. This suggest that the device can only be used for biosafety level 1 organisms. How the Authors can justify its use with eukaryotic cells? Also, appropriate procedures for the device assembly and decontamination should be explained.
  3. Figures 8 and 9 are reproduced from the previous publication. In the article, describing the original research, this is unacceptable. The citations of the previous work should be provided instead. Besides, it is not clear what does the empty panel under “enzymatic labelling” shows
  4. It would also be very useful to see higher magnification images showing that the cell structures (cytoskeleton, mitochondria, etc.) are not damaged during the procedure. This is especially important in the sample showing the isolated mitochondria.

Minor comments:

  1. Lines 185-187: “Although the same effect provides shortening of the time and/or the speed of centrifugation if dry arrangement is used, the wet arrangement is more reliable for the preparation of wet samples.” – The meaning unclear, please revise.
  2. On the University web page, the discussed device is referred to as CytoTrap and one can assume it is in the early stages of commercialization. It would make a lot of sense to use the same name in the publication.

Author Response

Thank you for all your valuable comments. Below find our replies.

1, Lines 204-208: “The subsequent experiments showed that even the higher efficiency of the cell capture was observed if an incubation with copper ions was performed after the formaldedyde fixation of cells. We found that the treatment of the captured cells on non-coated coverslips with 2-50 mM copper sulphate shortly after the formaldehyde fixation is sufficient for stabilization of cells if wet arrangement is used.”

– no data is provided to support this claim. Comparison of images with and without incubation with copper sulfate should be provided, quantification and statistics from repeated experiments are also required. Mechanisms leading to cell stabilization should be discussed.

We added the data supporting this finding to the text. In this respect, we added new Figures 5a and 5b showing the images of the cells prepared by wet centrifugation with and without the incubation with the copper ions (Figure 5a) and the cells prepared by dry centrifugation (Figure 5b). We also added the value of the ratio between the average number of cells calculated in the samples prepared with and without copper ions from three independent experiments.

The possible mechanism of such stabilisation was discussed.

2, It seems that all parts of the device come in contact with the biological sample. Moreover, during the centrifugation no lid is used, which carries a danger for aerosol formation and spills. This suggest that the device can only be used for biosafety level 1 organisms. How the Authors can justify its use with eukaryotic cells? Also, appropriate procedures for the device assembly and decontamination should be explained.

  • We added the note about the possibility to use the cap to the manuscript. In this respect, we added to the supporting data new figure (S1 Figure) showing the device with the cap. The device can be used with or without this cap which is put on the part of the insert rise above the lid. Depending on the solutions and biohazard features of the sample, the user can therefore decide to use or not to use it during centrifugation.
  • We added the S2 Figure showing the procedure of the CytoTrap assembly.
  • Concerning the decontamination, we added the information about materials from which the CytoTrap is manufactured (PTFE and silicone rubber) and also the temperature stability of these materials. In this respect, we also added the note that autoclaving can be used for decontamination of these parts.

3, Figures 8 and 9 are reproduced from the previous publication. In the article, describing the original research, this is unacceptable. The citations of the previous work should be provided instead. Besides, it is not clear what does the empty panel under “enzymatic labelling” shows

We removed Figures 8 and 9 already published in our previous article. We added new Figure 9, which shows the deposition of two bacterial strains on coverslips using CytoTrap. We only left the citation and short description concerning the original Figure 9 in the text.

4, It would also be very useful to see higher magnification images showing that the cell structures (cytoskeleton, mitochondria, etc.) are not damaged during the procedure. This is especially important in the sample showing the isolated mitochondria.

We replaced the Figure 8a with the figure with higher resolution to see more structural details like nucleoli or mitotic chromosomes. Anyway, we are aware the difficulties concerning the preparation of samples used for the high resolution by cytocentrifugation as the typical protocol includes the cell drying. It inevitably results into many artifacts. Even ethanol fixation is not convenient for such studies. As CytoTrap can be used for the preparation of formaldehyde-fixed samples by wet centrifugation, it can overcome some of these problems.

Concerning the sample with mitochondria, we used previously developed protocol of Frezza et al., 2007) for their isolation. According to the authors, this protocol makes it possible to perform some kinetic studies. On the other hand, we did not found any note about the level of the structure preservation. As the size of human mitochondria is around 0.5 mM, the control of the structural details would require embedding the sample in an embedding media and sample processing for electron microscopy. Although this is theoretically possible using the CytoTrap, its primary role we see in the preparation and staining of samples for light microscopy. On the other hand, formaldehyde fixation and substantially higher centrifugation forces than those used during sample processing by CytoTrap are commonly used during the sample preparation for electron microscopy studies. Therefore, we are convinced that CytoTrap-induced changes are fewer or comparable with the induced changes in such studies.

5, Lines 185-187: “Although the same effect provides shortening of the time and/or the speed of centrifugation if dry arrangement is used, the wet arrangement is more reliable for the preparation of wet samples.” – The meaning unclear, please revise.

We removed this sentence from the text.

6, On the University web page, the discussed device is referred to as CytoTrap and one can assume it is in the early stages of commercialization. It would make a lot of sense to use the same name in the publication.

We replaced the term device/apparatus by the term CytoTrap where it was appropriate.

Round 2

Reviewer 1 Report

The authors have done an outstanding job responding to my comments on the previous version of their manuscript. I am pleased to recommend that the manuscript be accepted for publication in its current form.

Reviewer 2 Report

The Authors have adequately addressed all the questions and I am happy to recommend the manuscript for publishing.